# Inkjet Printable and Self-Curable Disperse Dyes/P(St-BA-MAA) Nanosphere Inks for Both Hydrophilic and Hydrophobic Fabrics

**DOI:** 10.3390/polym10121402

**Published:** 2018-12-18

**Authors:** Yawei Song, Kuanjun Fang, Yanfei Ren, Zhiyuan Tang, Rongqing Wang, Weichao Chen, Ruyi Xie, Zhen Shi, Longyun Hao

**Affiliations:** 1Fiber Materials and Modern Textiles of the Growing Base for State Key Laboratory, Qingdao University, 308 Ningxia Road, Qingdao 266071, China; 15864737168@163.com (Y.S.); tjpuryf@126.com (Y.R.); 18363995943@163.com (Z.T.); 17864283938@163.com (R.W.); chenwc@qdu.edu.cn (W.C.); xry1228@126.com (R.X.); shizhen988@126.com (Z.S.); hly1978@163.com (L.H.); 2School of Textiles & Clothing, Qingdao University, 308 Ningxia Road, Qingdao 266071, China; 3Collaborative Innovation Center for Eco-Textiles of Shandong Province, 308 Ningxia Road, Qingdao 266071, China

**Keywords:** inkjet printing, disperse dyes, P(St-BA-MAA), color polymer nanospheres, stability, self-curable

## Abstract

Low-water-soluble disperse dyes possess a broad color gamut and good durability, but they need chemical or physical modification before being used in inks and can only be applied to several kinds of hydrophobic fabrics. In this work, disperse dyes/P(St-BA-MAA) nanospheres (known as DPN) absorbed by sodium nitrilotriacetate (known as NTA@DPN) were prepared and applied into ink formulations, which exhibited high dye fixation, long-term stability and self-curable ability without addition of any binder. Transmission electron microscopy (TEM) images showed the nanospheres have homogeneous core-shell spherical shape and the average diameter increased by 20.6 nm after coloration. X-ray diffraction (XRD), Fourier transform infrared spectrum (FTIR), and differential scanning calorimetry (DSC) measurements illustrated the interaction between dyes and nanospheres and indicated that the colored nanospheres contained both dye molecules and crystalline dyes. The Zeta potential and particle size measurements demonstrated that the dispersion stability was improved when sodium nitrilotriacetate (NTA) was absorbed onto DPN. The rheological behavior of the NTA@DPN inks was Newtonian and desired droplet formation was achieved at the viscosity of 4.23 mPa·s. Both hydrophilic cotton and hydrophobic polyester fabrics were cationic modified before used, which had an excellent image quality and desired rubbing fastness after inkjet printing. Scanning electron microscope (SEM) images showed NTA@DPN formed stable deposits on the surface of modified fibers and could self-cure to form continuous film coating on the fiber surface after being baked at 150 °C without addition of any binder.

## 1. Introduction

Inkjet printing has been proved to be a promising technology in textile printing [1]. Simultaneously, as the main consumable for inkjet printing, inkjet inks have received increasing attention from academia and industry [2,3]. In general, inks are classified by the type of colorants. For dye-based inks, the low-water-soluble disperse dyes have a broader color gamut and better durability than water-soluble dyes, which makes them more suitable for high color-performance applications [4]. However, up to now, the preparation and application of disperse dyes in aqueous inks are still considered in a dilemma. For example, surfactants or polymeric dispersants, or appropriate polymer encapsulation, is essential because of its low dispersion stability [5,6], but it may affect the interaction between dye molecules and fibers. Moreover, although it has such good properties, disperse dyes can only be applied to certain kinds of hydrophobic fabrics, not for any hydrophilic fabrics.

To overcome such limitations and to extend the application of disperse dyes, color polymer nanospheres based on disperse dyes are studied and show great potential in inkjet inks. Color polymer nanospheres combine dyes acting as an essential ingredient and suitable polymeric matrix, exhibiting good performance integrating the excellent chromatic properties and good processability [7]. To date, several methods on the preparation of color polymer nanospheres have been reported, such as solvent evaporation method and mini-emulsion polymerization [8]. Although these methods are effective for incorporating disperse dyes into polymer particles, it is difficult to obtain pure products in most cases owing to the polymeric stabilizer or surfactant added in rapid solvent evaporation method, and mini-emulsion polymerization method is time-consuming and complex, requiring control over numerous parameters such as monomer composition, reaction conditions, and emulsifier type [9,10]. In our previous work [11], a facile and environmental method was investigated in which poly(styrene-*co*-acrylic acid) nanosphere dispersions were directly colored with commercial disperse dyes and pure-colored nanosphere dispersions were obtained after a simple purification procedure.

Long-term stability and low tendency of clogging of printing nozzles are essential prerequisites for inks, especially in aqueous media, so it is crucial to prevent the association and agglomeration of nanospheres and enhance the stability of colored nanospheres [12]. According to DLVO theory (A theory about colloidal stability named after Derjaguin and Landau and Verwey and Overbeek), the stability of the polymer nanoparticle dispersion is governed by attractive energy and repulsive energy. The former is from van der Waals forces. The latter is due to solvation, electrostatic repulsion, and steric repulsion [13,14]. In this study, emulsifier-free emulsion polymerization prepared poly(styrene-butyl acrylate-methacrylic acid) (P(St-BA-MAA)) was chosen to be the polymer nanosphere matrix. Butyl acrylate (BA) is a kind of polar and soft monomer with a plenty of applications in emulsion polymerization to reduce the glass transition temperature [15]. Methacrylic acid (MAA) is a common functional monomer used in emulsifier-free polymerization to give nanospheres self-dispersibility [16,17]. To further improve the repulsive energy between nanospheres, non-ionic or ionic surfactants were used by many researchers [18,19,20]. However, in ink formulations, plenty of bubbles would be produced and rheological properties may be changed when so much non-ionic or ionic surfactant exists, which increases the risk of unfavorable inkjet droplet formation such as satellite droplets, drop splashing and fake clogging. Sodium nitrilotriacetate (NTA) is of particular interest from a chemical perspective because of the multiple stoichiometries and protonation levels when it coordinates metal ions [21,22]. It exhibits good low viscosity, low foaming ability and non-corrosive properties, all of which is suitable for ink formulation and has been rarely reported.

The objective of this study was to optimize materials and conditions for preparing disperse dyes/P(St-BA-MAA) nanospheres (known as DPN) inks for both hydrophilic and hydrophobic fabrics with high dye fixation, long-term stability, and self-curable ability without addition of any binder. P(St-BA-MAA) nanosphere dispersions were prepared with emulsifier-free polymerization and directly colored with commercial disperse dyes at 95 °C. Pure-colored nanospheres were obtained after a simple purification procedure. We determined the amounts of dyes fixed into the nanospheres. The colored nanospheres were characterized by X-ray diffraction (XRD), Fourier transform infrared spectrum (FTIR) analysis and differential scanning calorimetry (DSC) measurement. The colored nanosphere inks were printed on to cationic modified cotton and polyester fabrics, and the fiber morphology was observed by scanning electron microscope (SEM).

## 2. Materials and Methods

### 2.1. Materials

The commercial disperse dyes, C. I. disperse Red 60, C. I. disperse Blue 60, C. I. disperse Yellow 114, and disperse black ECT were supplied by Hongda Chemical Industrial Co., Ltd., Yanzhou, China, and used as received. Styrene (St), BA methacrylic acid (MAA) and ammonium persulfate (APS) were purchased from Shanghai Aibi Chemical Co., Ltd., Shanghai, China. These chemicals were purified according to standard procedures. Sodium nitrilotriacetate (NTA) powder was supplied by Ascend Performance Materials LLC., Houston, Texas, USA. 2,3-epoxypropyltrimethylammonium chloride (EPTAC), ethylene glycol (EG) and *N*,*N*-Dimethylformamide (DMF) were purchased from Qingdao Huadong Chemicals and Equipment Co., Ltd., Qingdao, China. Water was distilled before used in all the experiments. Cotton fabrics (C40 × 40, 133 × 72, Yu Yue Home Textiles Co., Ltd., Binzhou, China.) and polyester fabrics (75D × 150D, 91 × 63, Yu Yue Home Textiles Co., Ltd. Binzhou, China.) was used as substrate for the inkjet printing.

### 2.2. Synthesis of P(St-BA-MAA) Nanospheres

An emulsifier-free emulsion polymerization was employed to prepare P(St-BA-MAA) nanospheres using St, BA and MAA as monomers, and APS as the initiator [23,24]. Typically, 12.3 g of St and 1.2 g of BA were added to 70 mL of H_2_O. The mixture was transferred to a flask equipped with a Eurostar digital agitator (IKA Lab Technology, Guangzhou, China), a reflux condenser, a N_2_ tube, and a thermometer. The system was heated to 70 °C with N_2_ for 90 min followed by dropwise addition of 1.5 g of MAA and 0.2 g of APS dissolved in 25 mL of H_2_O over a 1.5 h period by a constant-pressure funnel. The system was maintained at 70 °C for 7 h with stirring at a rate of 350 rpm. After the reaction was completed, the product was filtered using a Buchner funnel to eliminate the larger particles.

### 2.3. Preparation of DPN Inks

A 0.2 g sample of C. I. disperse Red 60 (shown in Scheme 1) and 100 mL H_2_O were added to a beaker, followed by dropwise addition of 100 mL of the nanosphere dispersion under magnetic stirring. The pH of the mixture was adjusted to 9. Then, the mixture was transferred to a moderate shaking speed in a SHA-B shaker (Changzhou Guohua Electric Appliance Co., Ltd., Changzhou, China) and treated for 10 min at 50 °C. The system was subsequently heated to 95 °C and maintained the temperature for 60 min. Subsequently, the colored sample was taken out of the machine and cooled to room temperature [11].

Purified colored nanospheres were obtained from the dispersion via a series of steps. First, 10 mL of HCl solution (1.0 mol/L) was added to 50 mL of the colored sample under magnetic stirring, and then the colored nanospheres were separated by 3 cycles of centrifuging-washing with 0.1 mol/L HCl solution and 2 cycles with deionized water. Finally, purified colored nanosphere dispersion was transferred into evaporating dishes and dried at 60 °C. The colored nanosphere inks were prepared by dispersing the dried nanospheres into a mixture of pure water, EG and a non-ionic surfactant Surfynol 465 (S465). The surface tension and viscosity adjusted to the required values.

### 2.4. Droplet Formation Observation and Inkjet Printing

Droplet formation observation and inkjet printing was carried out using a droplet watcher (Hangzhou Fanjiang Electronic Technology Co., Ltd., Hangzhou, China) fitted with a Drop-on-Demand (DOD) inkjet printer: Galaxy JA 256/80 AAA inkjet printer (Fujifilm Dimatix, Inc., Santa Clara, CA, USA) and a drop-watch camera system which can record the drop characteristics when it is ejected from the nozzle.

The cotton and polyester were modified by EPTAC before being used. Firstly, 2 g of EPTAC and 0.5 g of NaOH powder was added to a beaker, and then 100 mL water was combined with stirring. Secondly, the cotton and polyester fabrics were fully impregnated with the liquid and padded with 80% pick-up by using a padding machine (Xiamen Rapid Co., LTD., Xiamen, China). Thirdly, the fabrics were transferred into an oven (Shanghai Yiheng Scientific Instrument Co., Ltd., Shanghai, China), and treated at 120 °C for 2 min. Finally, after being washed with pure water, the modified fabrics were dried at 50 °C.

The inkjet printing was conducted using the parameters Pixel 500 dpi, 5 pass, and subsequently baked at 150 °C for 2 min with a Minni thermo-350 baker (Roaches Company, Dewsbury, UK), and the rubbing fastness was tested with a dyeing color fastness meter (Laizhou Electronic Instrument Co., Ltd., Laizhou, China) according to Chinese national standard GB/T 3920-2008.

### 2.5. Sizes and Zeta Potentials of the Color Polymer Nanospheres

The nanosphere samples were diluted into the concentration of 0.2 g/L with deionized water. The average values of sizes and Zeta potentials of colored polymer nanospheres were determined by using a Nano ZS90 instrument (Malvern Panalytical Ltd, Malvern, UK) at 25 °C [25].

### 2.6. Observation by TEM

The colored nanosphere inks were ultrasonically diluted 1000-fold with deionized water and dropped onto Cu meshes. After drying under an infrared lamp, the morphology of nanospheres was determined by JEM-1200EX transmission electron microscopy (JEOL Ltd., Tokyo, Japan) [26,27].

The diameters of 100 different nanospheres on the TEM images were measured. The average diameter *D* was calculated according to the formula, $D=\sum_{i=1}^{n}d_{i}/n$, where *n* equals 100, and *d_i_* is the diameter of nanospheres *i*.

### 2.7. Observation by SEM

The printed cotton and polyester fabrics samples obtained above were dried at 50 °C for 20 min. Then, small pieces of samples were cut down and fixed onto the test bench with a conductive tape. The samples were sputter-coated with gold. The SEM pictures were taken using an S-4800 field-emission scanning electron microscope (Hitachi, Ltd., Tokyo, Japan).

### 2.8. Dye Content Measurements

A known weight of disperse dye was dissolved in DMF and the concentration was adjusted to 0.01 mg/ml. The colored nanosphere powder was also dissolved in DMF to achieve a dye concentration of approximately 0.1 mg/mL. The visible absorption spectrum of the DMF solution was measured using a double-beam UV-vis spectrophotometer (TU-1901, Beijing Purkinje General Instrument Co., Ltd., Beijing, China). The absorbance of the DMF solution was then measured at the maximum absorption wavelength of the dye. The maximum absorption wavelengths of C. I. disperse red 60, C. I. disperse blue 60, and C. I. disperse yellow 114 in DMF solution are 521, 656, and 442 nm, respectively. The dye content of the colored nanosphere was calculated using Lambert–Beer’s law [28]. According to the standard curve of disperse dyes, the amount of disperse dyes in the composite nanospheres was obtained. The dye content *C* (mg·g^−1^) in nanospheres can be calculated by Equation (1):
(1)$$C=\frac{m_{1}}{m_{0}-m_{1}}$$
where *m*_0_ is the mass of color polymer nanospheres and *m*_1_ is the mass of disperse dyes in composite nanospheres.

### 2.9. X-ray Diffraction Measurements (XRD)

A certain amount of colored nanosphere dispersion was transferred into an evaporating dish and dried at 60 °C, then milled into powder. X-ray diffraction measurements were carried out with D8 ADVANCE X-ray diffractometer (BRUKER AXS Instrument Co., Bruker, Germany). The diffraction angle is 0~40° [29]. The results were analyzed by JADE 5 software (Materials Data, Inc., Livermore, California, USA).

### 2.10. Fourier Transform Infrared (FTIR) Analysis

The Fourier transform infrared spectrums of the disperse dye, P(St-BA-MAA) nanospheres and colored nanospheres powder were obtained by a TENSPR37 spectrometer (Bruker Co., Ltd., Hamburg, Germany). The scanning range was set at 400~4000 cm^−1^ with 1 cm^−1^ resolution [30].

## 3. Results and Discussion

### 3.1. Morphology of the DPN

TEM images of the nanospheres before and after coloration are shown in Figure 1. As shown in Figure 1a, the P(St-BA-MAA) nanospheres synthesized by the emulsifier-free emulsion polymerization had homogeneous core-shell spherical shape with an average size of 374.1 nm. In Figure 1b, the morphology of disperse dyes/P(St-BA-MAA) nanospheres (named as DPN) did not change evidently after coloration, but the average diameter increased to 394.7 nm, 20.6 nm larger than the uncolored ones resulting from the interaction between dyes and nanospheres [11,28].

### 3.2. The Interaction between Disperse Dyes and P(St-BA-MAA) Nanospheres

The influence of dyeing pH on dye content is shown in Figure 2a. It is clear that the dye content increased from 24.41 mg/g to 35.3 mg/g with the increase of the bath pH value from 4.11 to 9.61. X-ray diffraction analysis of the nanosphere powder (Figure 2b) showed that the disperse Red 60 powder (black line) exhibited multi-diffraction peaks in the angle range of 0–35°, exhibiting the multi-crystalline structure properties of disperse dyes [31,32]. No significant diffraction peaks appeared in P(St-BA-MAA) powders (Green line). After the coloration (blue line and red line), in the angle range of 25–35°, the diffraction peaks appeared apparently, but no change took place in the range of 0–25°, revealing that the colored nanospheres contain crystalline dyes. There was only a single weak diffraction peak at 31.62° with the crystallinity of 1.12% when the nanospheres were dyed at pH 4 (blue line), due to the low dye content of the nanospheres (Figure 2a). In comparison, the colored nanospheres dyed at pH 9 (red line) showed many more diffraction peaks at 26.15°, 27.42°, and 31.62° with a higher crystallinity of 17.81%. FTIR spectrums of the disperse dye (black line), P(St-BA-MAA) nanospheres (green line) and colored nanospheres (red line) are shown in Figure 2c. The characteristic peaks of 2930 and 2844 cm^−1^ of all samples could be ascribed to the stretching vibration of C-H. For both uncolored and colored nanosphere samples, the characteristic peak around 3020 cm^−1^ is assigned to stretching vibration of =C–H. The peak of 1723 cm^−1^ was due to the stretching vibration of C=O of acrylate. The stretching vibration of –COO^−^ of MAA were at 1598 and 1380 cm^−1^. Particularly, in the spectrums of colored nanospheres, a new characteristic band appeared at 1567 cm^−1^ resulting from the bending vibration of N–H of disperse Red 60 molecules, indicating the presence of the dyes in the nanosphere. From the DSC curves of the uncolored and colored nanosphere powder dyed at pH 9 (Figure 2d,e), it could be seen that the glass transition temperature (*T*_g_) of the colored nanospheres increased from 92.3 °C for the uncolored nanospheres to 102.8 °C, increased by 10.5 °C after the coloration. This result indicated the mobility of the P(St-BA-AA) chains decreased due to the coloration of disperse dyes.

When the bath pH is under 5, only a few of the –COOH groups on the nanosphere surface can be ionized into –COO^−^. The macromolecular chains are tightly entangled through inter-macromolecule hydrogen bonds and Van der Waals force, so it is difficult for dye molecules to enter the inner part of the nanosphere. In our previous study [33], it has been proved that the –OH and –NH_2_ of disperse Red 60 could form hydrogen bonds with the carboxyl groups on the surface of the nanosphere (Figure 3a). When the bath pH increased to basic conditions (pH over 9), more of the –COOH groups on the surface were converted into –COO^−^ and the interactions among macromolecular chains were damaged, which resulted in an increase of the distance among macromolecular chains. Therefore, more dye molecules can easily diffuse into the nanosphere, which may form new hydrogen bonds with carbonyl groups from BA (Figure 3b) and establish Van der Waals force with the macromolecules. With the increase of dye content in nanosphere, the disperse dyes are easy to aggregate to form dye crystals. Therefore, as shown in Figure 3c, the colored P(St-BA-MAA) nanospheres not only contain dye molecules but also contain crystalline dyes.

### 3.3. The Stability of DPN Dispersions

Chelating agents are usually used to treat hard water, and have been included in formulations to prevent thickening and/or gelling [34]. Sodium nitrilotriacetate (NTA, showed in Scheme 1), a usual chelating agent, was investigated in this study to improve the stability of DPN. NTA powder was added into the DPN dispersion at pH 8 by stirring for two hours. To reveal the adsorption of NTA onto DPN more intuitively, TEM images were determined, as shown in Figure 4. The colored nanospheres exhibit clear spherical structure (Figure 4a). Comparatively, in Figure 4b, it can be clearly observed that a large amount of NTA have been absorbed onto the surface of DPN (known as NTA@DPN) and formed a shell layer.

Zeta potential is an important parameter to measure the stability of colloidal dispersions, which are determined by the surface electrical property and quantity of surface charges. The effects of pH on Zeta potential of the colored nanosphere dispersion and NTA@DPN dispersion are shown in Figure 4c. The absolute values of negative Zeta potentials for both colored nanospheres and NTA@DPN increased when increasing the bath pH values, indicating that more –COOH groups on the surface of nanospheres ionized into –COO^−^ form, meaning that the electrostatic repulsive force between the nanospheres increased. In comparison, the absolute values of Zeta potentials for NTA@DPN are higher than those of the colored nanospheres. It proves that NTA was absorbed onto the surface of colored nanospheres. However, further increasing the pH to nearly 9, the Zeta potential of both samples reached about the same value. Figure 4d shows the NTA@DPN dispersions stability vs. time at pH about 8. It demonstrates that the particle size distributions of NTA@DPN dispersions only have a slight change over 15 days and the DPN dispersion changed significantly. 

### 3.4. Rheological Behavior and Droplet Formations

The rheological properties of the colored nanosphere dispersion have great influence on the droplet formations [35,36]. Flow curves of NTA@DPN inks with different amounts of EG are shown in Figure 5a. As it demonstrates, the shear stress varied linearly with the changes of shear rate both for water and all ink samples. It indicates rheological behavior of the colored nanosphere inks is Newtonian, with viscosities ranging from 1.5 to 5 mPa·s. Thus, the formulations include EG as a rheological modifier are suitable for DOD inkjet printing to avoid secondary phenomena, which usually happens for non-Newtonian liquid, such as sedimentation and dripping from nozzles [35].

Suitable viscosity and surface tension for the DOD inkjet printing are the crucial properties to obtain proper droplet formations, which have great effects on image quality. Figure 5b–e are the droplet formations of the NTA@DPN inks with different formulations discussed in Figure 5a. The viscosities are 1.82 (Figure 5b), 2.79 (Figure 5c), 3.87 (Figure 5d) and 4.23 (Figure 5e) mPa·s, respectively. The surface tension is 30~35 mN/m according to the requirements of inkjet printing and the constant drive voltage was 50 V. A satellite droplet is undesirable because it is far more readily misdirected and can thereby degrade the printing resolution [37]. From Figure 5b to Figure 5d, satellite droplets formed significantly. That is because, when the viscosity is low, the viscous force cannot prevent the separation of liquid layers with different flow speeds, so satellite droplets formed [37,38]. When the viscosity continues to increase, viscous force is improved, and the “tail” can shrink into the parent droplets. Then, it is difficult to form satellite droplets, and desired droplet formations are finally achieved (Figure 5e).

### 3.5. Self-Curing on Both Hydrophilic and Hydrophobic Fabrics

The NTA@DPN inks were employed in the inkjet printing of hydrophilic cotton fabrics and hydrophobic polyester fabrics, which were modified by EPTAC to deposit NTA@DPN on the fiber surface to avoid color bleeding. The SEM images of NTA@DPN printed fabrics were exhibited in Figure 6. It shows that fiber morphology did not change after modification (Figure 6a,d) and NTA@DPN were firmly absorbed on the modified fiber surfaces (Figure 6b,e). However, the distribution on cationic modified polyester fibers is more concentrated than that of cotton fibers. This phenomenon is due to the different modification mechanisms for cotton and polyester fabrics. EPTAC can react with –OH on the cotton fiber surface, so the absorption sites are distributed throughout the fiber surface. In comparison, EPTAC can only be adsorbed onto the polyester fiber surface through hydrophobic forces, which lacks uniformity. Therefore, the distribution conditions of NTA@DPN on two kinds of fabrics were different, accordingly.

After baking, as shown in Figure 6c,f, the spherical morphology of nanospheres has changed and formed a continuous film coating on the fiber surface, i.e., the nanospheres have self-cured. Further experiments with blue, yellow, and black NTA@DPN inks (Figure 7c) are also prepared with the disperse dyes (Appendix A) and printed onto the cationic modified cotton and polyester fabrics. The printed images are exhibited in Figure 7a,b. The CIE (Commission International Eclairage) Lab color values are measured [39] and summarized in Appendix A. At the same time, the rubbing fastness was measured and shown in Figure 7d,e. Both hydrophilic cotton and hydrophobic polyester fabrics have good color performance and desired rubbing fastness after self-curing.

## 4. Conclusions

The disperse dyes/P(St-BA-MAA) nanospheres (known as DPN) were prepared through surfactant-free emulsion polymerization and coloring by commercial disperse dyes. The nanospheres exhibited homogeneous spherical structure by TEM observation, and the average particle size of the nanospheres increased by about 20.6 nm after coloration. With the increase of bath pH in the range of 4~9, the dye content increased. X-ray diffraction analysis of the nanospheres dyed at pH 9 showed diffraction peaks at 26.15°, 27.42°, and 31.62° with a crystallinity of 17.81%, indicating that the colored nanospheres contained both dye molecules and crystalline dyes. From the FTIR spectrum analysis, it was demonstrated that the disperse dyes were present in the nanosphere. DSC measurements indicated the *T*_g_ increased by 10.5 °C after the coloration. Further improved stability was obtained after NTA was absorbed onto the surface of the DPN. The slight changes of particle size of NTA@DPN dispersion indicated the the dispersion system has a long-term stability at a mild pH. The rheological behavior of the NTA@DPN inks was Newtonian with viscosities ranging from 1.5 to 5 mPa·s. Desired droplet formation was achieved for the viscosity 4.23 mPa·s. 

The NTA@DPN inks were employed in the inkjet printing on cationic modified hydrophilic cotton fabrics and hydrophobic polyester fabrics. The SEM images showed that NTA@DPN could be absorbed onto the surface of cationic modified cotton and polyester fibers and the nanospheres could self-cure after baking at 150 °C. Both hydrophilic cotton and hydrophobic polyester fabrics had an excellent image quality and desired rubbing fastness after the inkjet printing of four-color NTA@DPN inks.

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
