# Peer review of "Inkjet Printable and Self-Curable Disperse Dyes/P(St-BA-MAA) Nanosphere Inks for Both Hydrophilic and Hydrophobic Fabrics"

_polymers, 2018, doi:10.3390/polym10121402_

Round 1

Reviewer 1 Report

1) Please specify more clearly (points 2.6 and 2.7) how the material for TEM and SEM analysis was prepared, how was the material obtained from the suspension?

2) The work should be enriched with the results of the FT-IR analysis, which could confirm or enrich the interpretation of the mechanism.

Author Response

Dear Reviewer:

Thank you for the comments of our manuscript (Manuscript ID: polymers-402183). These comments are all valuable and very helpful for revising and improving our paper. We have studied comments carefully and have made corrections. Revised portions are marked in red in the paper. The main corrections in the paper and the responses to the comments are as follows:

Comment 1: Is the research design appropriate? Can be improved.

Response: Thanks for your suggestions. The FTIR analysis was carried out to enrich the interpretation of the mechanism, and showed in Figure 2c (Page 6).

Comment 2: Are the methods adequately described? Can be improved.

Response: Thanks for your suggestions. All methods have been described more clearly in the paper.

Comment 3: Please specify more clearly (points 2.6 and 2.7) how the material for TEM and SEM analysis was prepared, how was the material obtained from the suspension?

Response: Thanks for your suggestions. The measurements of TEM and SEM (Page 4, line 146) have been specified clearly about the material preparation, and the purified method in point 2.3 (Page 3, line 123) was revised to show how the material was obtained from the suspension.

Comment 4: The work should be enriched with the results of the FT-IR analysis, which could confirm or enrich the interpretation of the mechanism.

Response: Thanks for your suggestions. We measure the FTIR spectrums of according to the Reviewer’s suggestion and as showed in Figure 2c. The analysis of the spectrums was showed in point 3.2 (Page 6, line 213). The measurement method was showed in point 2.10 (Page 4, line 177). The results of FTIR analysis have been added to Abstract, Introduction and Conclusions. As showed in revised manuscript.

Reviewer 2 Report

The following comments should be considered in the revised manuscript,

 page 4, 2.8. Dye content measurements : It is required to provide the value of the exact maximum absorption wavelength for individual three disperse dye.

2. page 6 : In the paragraph of "the glass transition temperature (Tg) of the color P(St-219 BA-AA) nanospheres increased from 92.3 °C to 102.8 °C, which increased by 2.98 °C", 2.98  °C should be corrected to 10.5 °C.

page 7 :

   1) In the paragraph of "When the bath pH was under 6.33, the -COOH groups on the surface  were partially converted into -COO-~", the possibility of the ionization from free carboxylic acid into carboxylate ion should be higher at basic conditions(pH over 9) rather than weakly acidic conditions(pH under 5). Therefore it is required to change this paragraph to be resonable discussions.

   2) At the same reason, the interactions of hydrogen bonds between dye molecule and the carboxylic acid groups to be increased at acidic conditions. It is required to explain why the dye content was increased by the pH was changed to be more alkaline.  

Author Response

Dear Reviewer:

Thank you for the comments of our manuscript (Manuscript ID: polymers-402183). These comments are all valuable and very helpful for revising and improving our paper. We have studied comments carefully and have made corrections. Revised portions are marked in red in the paper. The main corrections in the paper and the responses to the comments are as follows:

Comment 1: Moderate English changes required

Response: Thanks for your suggestions, the language of all pages has been carefully corrected and marked in red in the paper.

Comment 2: Is the research design appropriate? Can be improved.

Response: Thanks for your suggestions. To enrich the interpretation of the mechanism, FTIR analysis was carried out, and showed in Figure 2c (Page 6, line 194).

Comment 3: Are the methods adequately described? Can be improved.

Response: Thanks for your suggestions. All methods have been described more clearly in the paper.

Comment 4: Are the results clearly presented? Must be improved.

Response: Thanks for your suggestions. The results and discussions in the paper have been corrected again to make it more reasonable in the paper.

Comment 5: Are the conclusions supported by the results? Can be improved.    

Response: Thank you for the suggestions. The conclusions were revised according to the results (Page 11, line 327).

Comment 6: page 4, 2.8. Dye content measurements: It is required to provide the value of the exact maximum absorption wavelength for individual three disperse dye.

Response: Thank you for the suggestions. The point 2.8 have been re-written and the maximum absorption wavelengths of C. I. disperse red 60, C. I. disperse blue 60 and C. I. disperse yellow 114 were added in the measurement (Page 4, line 164).

Comment 7: page 6: In the paragraph of "the glass transition temperature (Tg) of the color P(St-BA-MAA) nanospheres increased from 92.3 °C to 102.8 °C, which increased by 2.98 °C", 2.98 °C should be corrected to 10.5 °C.

Response: Thanks for your suggestions. The results of glass transition temperature (Tg) of the color P(St-BA-AA) nanospheres have been corrected (Page 6, line 224).

Comment 8: page 7 :1) In the paragraph of "When the bath pH was under 6.33, the -COOH groups on the surface were partially converted into -COO-", the possibility of the ionization from free carboxylic acid into carboxylate ion should be higher at basic conditions (pH over 9) rather than weakly acidic conditions (pH under 5). Therefore, it is required to change this paragraph to be reasonable discussions.

2) At the same reason, the interactions of hydrogen bonds between dye molecule and the carboxylic acid groups to be increased at acidic conditions. It is required to explain why the dye content was increased by the pH was changed to be more alkaline. 

Response: Thank you for the suggestions. We have re-written the discussions and explain the influence of pH on dye content (Page 7, line 231).